# Combining CDK4/6 inhibitors ribociclib and palbociclib with cytotoxic agents does not enhance cytotoxicity

Dan Jin◯⊙, Nguyen Tran¤⊙, Nagheme Thomas, David D. Tran◯*

Division of Neuro-Oncology, Lillian S. Wells Department of Neurosurgery, Preston A. Wells, Jr. Center for Brain Tumor Therapy, University of Florida College of Medicine, Gainesville, FL, United States of America

⊙ These authors contributed equally to this work.
¤ Current address: Lake Erie College of Osteopathic Medicine, Erie, PA, United States of America
* David.Tran@neurosurgery.ufl.edu

**Data Availability Statement:** All relevant data are within the manuscript and its Supporting Information files.

**Funding:** This work was supported by a preclinical grant from Novartis, Inc. and a grant from the

## Abstract

Cyclin-dependent kinases 4 and 6 (CDK4/6) play critical roles in the $G_1$ to S checkpoint of the cell cycle and have been shown to be overactive in several human cancers. Small-molecule inhibitors of CDK4/6 have demonstrated significant efficacy against many solid tumors. Since CDK4/6 inhibition is thought to induce cell cycle arrest at the G1/S checkpoint, much interest has been focused on combining CDK4/6 inhibitors with cytotoxic agents active against the S or M phase of the cell cycle to enhance therapeutic efficacy. However, it remains unclear how best to combine these two classes of drugs to avoid their potentially antagonistic effects. Here, we test various combinations of highly selective and potent CDK4/6 inhibitors with commonly used cytotoxic drugs in several cancer cell lines derived from lung, breast and brain cancers, for their cell-killing effects as compared to monotherapy. All combinations, either concurrent or sequential, failed to enhance cell-killing effects. Importantly, in certain schedules, especially pre-treatment with a CDK4/6 inhibitor, combining these drugs resulted in reduced cytotoxicity of cytotoxic agents. These findings urge cautions when combining these two classes of agents in clinical settings.

## Introduction

The cell cycle is comprised of four distinct phases, S, M, G1 and G2 gap phases. Progression to each phase is tightly regulated by different pairs of cyclin-CDK complexes [1], which monitor the order, integrity and fidelity of major events of the cell cycle, e.g. adequate mitotic signals for the G1-S checkpoint and faithful duplication and repair of DNA for the G2-M checkpoint [2]. Deregulation of these processes is a hallmark of oncogenesis [3]. Among cyclin-CDK complexes, those at the G1 phase are particularly important because they determine the ultimate commitment either to enter the cell cycle or to exit it entirely and remain quiescent. As a consequence, gain of function of the G1-S checkpoint's cyclin-CDK complexes has been shown to be a major driver in a large number of human cancers [4].

The D-type cyclins and their binding partners CDK4/6 are important regulators in the G1 to S checkpoint. Cyclin D expression is upregulated in response to external signals including

National Institutes of Health (1K08CA160824) (D. D.T). The funders had no role in the study design, data collection and analysis, decision to publish or preparation of the manuscript.

**Competing interests:** I have read the journal's policy and the authors of this manuscript have the following competing interests: D.D.T reports grants and personal fees from Novocure, personal fees from Monteris, grants from Merck, grants from Novartis, grants from VBL, grants from Stemline, grants from NWBiotechnology, grants from Lacerta Therapeutics, grants from Tocagen, outside the submitted work; In addition, D.D.T has a patent Master regulators of GBM pending, a patent GeneRep and nSCORE pending, a patent Methods of Cancer Screening and Monitoring pending, a patent AAV variants targeting GBM pending, a patent Mechanism of resistance to TTFields in GBM pending, a patent Methods for Targeted Treatment and Prediction of Patient Survival in Cancer pending, and a patent Chemotherapeutic Resensitization of Disseminated Cancer Stem Cells through Reactivation by P38 Inhibition and IL-6 Inhibition - Chemotherapeutic Methods pending. However, this does not alter our adherence to PLOS ONE policies on sharing data and materials.

stimulatory mitogens, inhibitory cytokines, differentiation factors and cell-to-cell contact. Cyclin D then binds to and activates CDK4/6. Activated cyclin D-CDK4/6 complex in turn phosphorylates the retinoblastoma protein, RB1, causing release of the E2F family of transcriptional factors normally sequestered by hypophosphorylated RB1 [5, 6]. Released E2F is then translocated into the nucleus where it is necessary for the transcription of genes responsible for cell-cycle progression. Consequently, overexpression of the cyclin D-CDK4/6 complex leads to uncontrollable growth, supporting the premise that inhibition of this complex has therapeutic potential as a cancer treatment. Studies using CDK4/6 inhibitors in cancer cell lines showed rapid cell cycle arrest at the G1/S checkpoint followed by senescence and, in some cases, apoptosis [7, 8]. Currently, three orally bioavailable small-molecule CDK4/6 inhibitors, including ribociclib (Kisquali or LEE011; Novartis), palbociclib (Ibrance or PD-0332991; Pfizer), and abemaciclib (Verzenio or LY-2835219; Eli Lilly) have received regulatory approval in combination with hormonal therapy for treatment of patients with metastatic hormone receptor (HR)-positive, Her2-negative breast cancer [9–13]. Compared to the first two generations of CDK inhibitors, these three third generation CDK inhibitors are highly selective CDK4/6 inhibitors with more specific interactions with non-conserved elements of the ATP-binding pocket of the kinase domain [14], resulting in significantly less off-target effects and fewer dose-limiting toxicities [15]. Of these three agents, ribociclib was also found to be significantly more selective towards CDK4 and CDK6 than the other two [14, 16], and will be the main focus of the current study.

Although single-agent activity has been reported for CDK4/6 inhibitors, combinations with other drugs has demonstrated enhanced anti-cancer efficacy, including combination of ribociclib with an aromatase inhibitor as approved as a first-line treatment for postmenopausal women with metastatic HR+/HER2- breast cancer [17]. Ribociclib combined with dexamethasone also resulted in increased cell death in B and T cell acute lymphoblastic leukemia [18, 19]. However, combinations of ribociclib and cytotoxic drugs have not been extensively studied. In theory, CDK4/6 inhibitors should be combined with cytotoxic agents that target the S or M phase of the cell cycle in order to capture tumor cells that may have escaped the cytostatic effects of CDK4/6 inhibition. However, recent studies indicated that exposure of RB1-intact breast cancer cells to a CDK4/6 inhibitor continuously prior to exposure to cytotoxic agents, such as doxorubicin and carboplatin, significantly reduced their cytotoxicity [20, 21]. This observation was in part expected due to the positive correlation between cytotoxicity of many cytotoxic drugs and the rate of cell division. In contrary, in certain ovarian cancer cell lines, additive or synergistic interactions were observed when a CDK4/6 inhibitor was given concomitantly with either carboplatin or paclitaxel [22]. However, another report suggested an antagonizing effect when a CDK4/6 inhibitor was combined concurrently with an anthracycline-based regimen in the triple-negative breast cancer cell line MDA-MB-231 [20]. In addition, due to the different phases of the cell cycle at which these two classes of drugs are active, whether a short exposure to one class of drugs will enhance response to the other, and if so, in what sequence, have not been rigorously addressed. Therefore, it remains unclear how to optimally combine these two classes of drugs to achieve additive or synergistic effects, while minimizing antagonism.

Here we tested various dosing schedules (e.g. concurrent versus sequential and continuous versus transient) of combinations of ribociclib and commonly used cytotoxic drugs in several cancer cell lines derived from human glioblastoma (GBM) and breast and non-small lung cancers to determine whether additive or synergistic cytotoxic effects can be realized when these two classes of drugs are combined. No additive or synergistic killing effects were observed in any of these dosing schedules. This lack of improved cytotoxicity appears to be a class effect as similar results were observed with palbociclib. Importantly, timed pre-treatment with

ribociclib to synchronize cells at the G1/S checkpoint followed by release and then exposure with cytotoxic drugs consistently resulted in a significant reduction in cytotoxicity. Our data indicate that until further investigation into this apparent antagonism is completed, cautions are warranted when combining these two classes of drugs in clinical settings.

## Materials and methods

### Cell lines, tissue culture and drug reagents

LN428 and LN308 were previously established from human GBM samples [23]. The human lung cancer cell line A549 and human breast cancer cell line MDA-MB-231 were obtained from ATCC (Cat. # CCL-185 and HTB-26). All cell lines were cultured in DMEM medium (Corning, 10013CV) supplemented with 10% fetal bovine serum (sigma, F0926),1% penicillin and 1% streptomycin, at 37°C with 5% $CO_2$ supplementation. Clinical grade ribociclib (LEE011) was obtained from Novartis, Inc. Carmustine (C0400), carboplatin (C2538), temozolomide (T2577), etoposide (E1383), irinotecan (I1406) were from Sigma-Aldrich; paclitaxel or taxol (S1150) and palbociclib (S1116) were from Selleck Chemicals.

### $IC_{50}$ determination and drug combination treatment

Cells were seeded for 24 hours before drug treatment in 96-well plates at a density of 1000 cells/well for LN428 and A549 and 2000 cells/well for LN308 and MDA-MB-231. For $IC_{50}$ determination of all drugs, cells were treated for 3 days with increasing drug concentrations. Drugs were replenished every 24 hrs. For treatment with drug combinations, cells were treated with different drugs at their $IC_{50}$ concentration as indicated. For synchronization drug combination treatment, cells were seeded in 6-well plates and treated with either ribociclib at $IC_{50}$ concentration or media alone for either 1 or 5 days. Subsequently, equal numbers of ribociclib or media treated cells were re-seeded in 96-well plates for 24 hours before cytotoxic agents were added for the next 3 days. Drugs were replenished every 24 hrs.

At the conclusion of treatment, cell viability was determined using Calcein AM reagent (Invitrogen, C3100MP) following the manufacturer's protocol [24]. Briefly, cells were washed once with DPBS and incubated in 4μM Calcein AM solution for 45 minutes at room temperature. Live cells were then enumerated, or the coverage area of live cells measured as percentage of the total area of the well in cases where cells grow as adherent clusters, using a Spectramax i3X plate reader. Dose-response inhibitory curves were generated and $IC_{50}$ determined using the statistical software Prism.

For comparison of more than two groups drug combinations, one-way ANOVA analysis was applied assuming Gaussian distribution of the dataset and using Tukey test to correct for multiple comparisons. For two groups comparison, unpaired, two-tailed student T-test was applied assuming both groups of treatment having the same standard of deviation.

### Cell cycle analysis

Cellular DNA content assay was performed as previously described [25]. Briefly, cells were seeded in 12-well plates such that on the day of DNA content determination the maximal confluency was 70–80%. Cells were treated with ribociclib at $IC_{50}$ concentration for 0 to 5 days starting at 24 hours after seeding. Ribociclib was replenished every 24 hrs. Cells were trypsinized, washed in DPBS, fixed in 70% ethanol at 4°C for 2 hours, and stained with propidium iodide (PI) (0.1% TritonX-100, 100μg/ml RNase, 10μg/ml PI) at 37°C for 15 minutes. Cellular DNA content was determined by flow cytometer and analyzed by FlowJo.

## BrdU incorporation assay

BrdU was performed as previously described [25]. Briefly, $3 \times 10^5$ cells were seeded in 6-well plate for LN428 and A549. One day after seeding, cells were treated with ribociclib at $IC_{50}$ concentration for 1 day, followed by 10μM BrdU containing media for 0, 2 or 4 hrs. Treated cells were then washed twice with DPBS, fixed in ice cold 70% ethanol, permeabilized in 1 M HCL solution containing 0.5% TritonX-100 for 30 minutes followed by 0.1 M $Na_2B_4O_7$, pH 8.5 for 2 minutes at room temperature, stained with FITC-labeled anti-BrdU monoclonal antibody (Biolegend, Cat# 364103) at 4˚C overnight, counterstained with 1 mg/ml propidium iodide, and analyzed by flow cytometry.

# Results

## Combining ribociclib with cytotoxic drugs does not increase cytotoxicity

CDK4/6 inhibitors are expected to have cytostatic effects by causing cell cycle arrest at the G1/S checkpoint. As a result, much effort has been focused on identifying optimal combinations of CDK4/6 inhibitors and other anti-cancer therapeutics. Combinations of CDK4/6 inhibitors and anti-hormonal therapy have demonstrated significant efficacy against HR-positive breast cancer [26–29]. However, it has proven more challenging to combine CDK4/6 inhibitors with cytotoxic drugs despite the fact that cytotoxic chemotherapy is much more widely used as anti-cancer treatment. This is due in large part to their distinct and potentially counteracting mechanisms of action, with some earlier reports demonstrating contradicting observations [19, 30, 31].

Based on where each of these two classes of drugs works during the cell cycle, we hypothesize that two potential dosing schedules may produce enhanced cell killing effects: 1) initial exposure of cycling tumor cells to a CDK4/6 inhibitor synchronizes cells at the G1 phase. Upon release from CDK4/6 inhibition, G1 synchronized cells that have not undergone irreversible cellular senescence will enter the S phase simultaneously when they are predicted to be maximally susceptible to cytotoxic drugs; or alternatively, 2) tumor cells treated first with a cytotoxic drug active at either the S or M phase will lead to arrest at either the G2/M or M checkpoint, respectively, and eventually to apoptosis. Those that can partially repair the cytotoxic insults and escape these checkpoints will enter the G1 phase where they may become susceptible to CDK4/6 inhibition. The presence of partially repaired DNA damage from prior exposure to cytotoxic drugs increases the rate that cells in G1 arrest due to CDK4/6 inhibition may undergo senescence or apoptosis. If either of these hypotheses is correct, combinations given in one of these sequential schedules are predicted to have at least an additive, if not synergistic, cytotoxic effect as compared to the concurrent schedule or monotherapy.

First, we determined the half inhibitory concentrations ($IC_{50}$) of ribociclib and six commonly used cytotoxic agents: two alkylating agents (temozolomide and carmustine) [32–34], the DNA cross-linker carboplatin [35, 36], the topoisomerase I and II inhibitors irinotecan [37] and etoposide [38], respectively, and the microtubule-stabilizing agent paclitaxel (Taxol) [36, 39] in four established human cancer cell lines including the two GBM lines LN428 and LN308 [23], the triple negative breast cancer line MDA-MB-231, and the lung adenocarcinoma line A549 (Fig 1A and S1A–S1G Fig). To reflect standard clinical usage, GBM cells were treated with temozolomide, carmustine, carboplatin, irinotecan, and etoposide, while lung and breast cancer cells with carboplatin and paclitaxel, respectively. Next, to test the above hypotheses, equal numbers of cancer cells were plated and treated with these two classes of drugs combined sequentially either with ribociclib first for 2 days followed immediately by 2 days of a cytotoxic drug or vice versa, and the number of remaining live cells determined at the end of

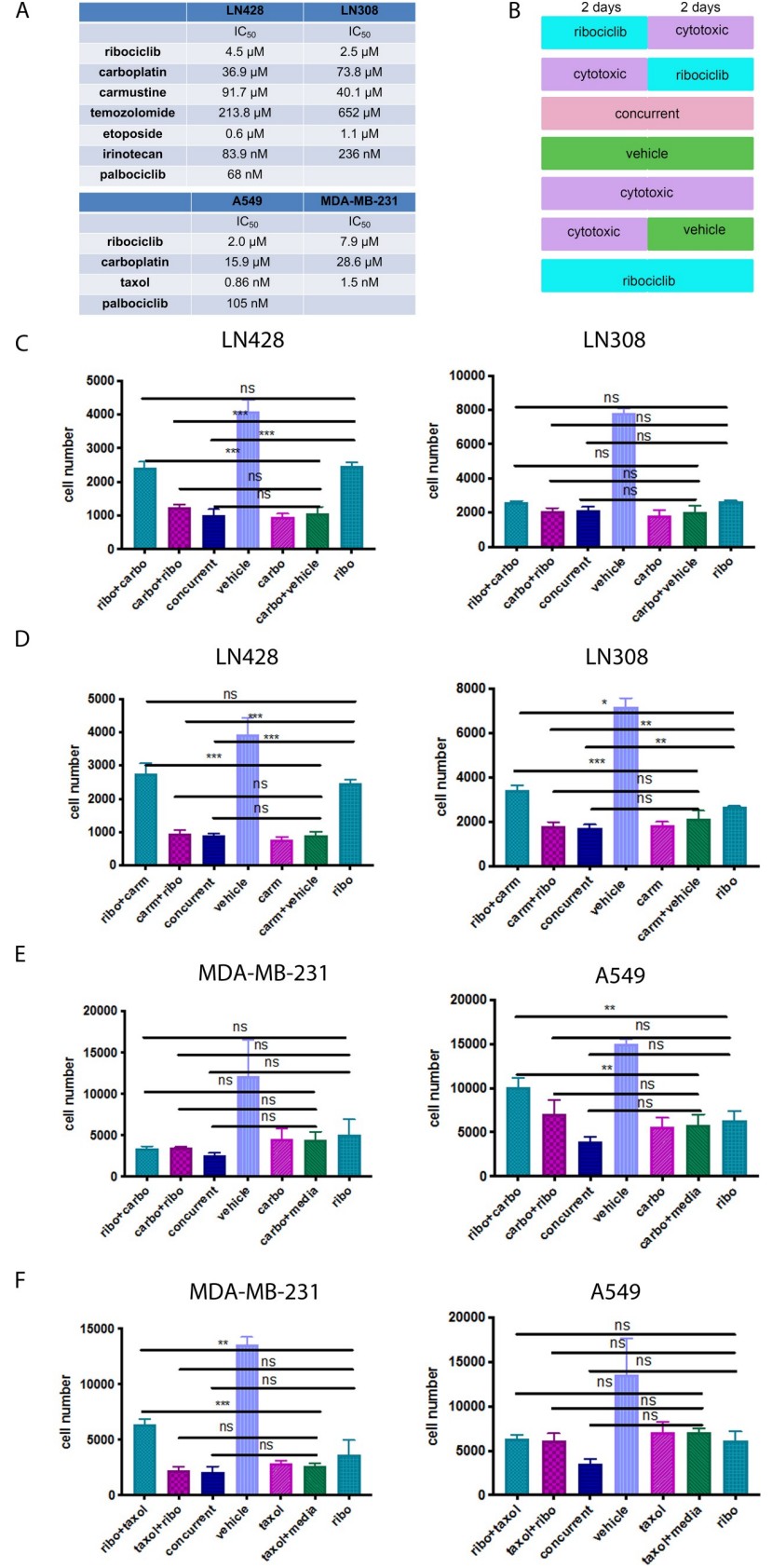

**Fig 1. Combining ribociclib with cytotoxic drugs did not increase cytotoxicity.** (A) Table of average $IC_{50}$ of three independent $IC_{50}$ determinations of indicated cytotoxic drugs in indicated cell lines. (B) Different dosing schedules of combinations of ribociclib and cytotoxic drugs. (C-F) Graphs of representative cytotoxicity assay of 3 independent repeats of the various combinations of ribociclib and indicated cytotoxic drugs as shown in B at the $IC_{50}$ concentration for each drug in LN428 and LN308 cells (C-D) and MDA-MB-231 and A549 cells (E-F). All values are numbers of live cells remaining in culture at the end of treatment and presented as mean (SD). P-value was calculated by one-way ANOVA: *, p<0.033; **, p <0.02; ***, p < 0.001.

treatment using the live cell fluorescence dye calcein [24]. For comparisons, the same cell lines were treated for a total of 2 days with a cytotoxic drug alone followed by 2 days of the vehicle or 4 days with either both drug classes concurrently or a cytotoxic drug alone or ribociclib alone (Fig 1B). Although concurrent combinations of ribociclib with etoposide and paclitaxel showed an additive effect in LN308 (S1I Fig) and A549 cells (Fig 1F), respectively, compared to cytotoxic drug or ribociclib alone, this additive effect was not observed in other cell lines or other cytotoxic drugs (Fig 1C–1F and S1H–S1J Fig). Sequential schedules of these two classes of drugs in all four cell lines failed to increase cytotoxicity as compared to treatment with cytotoxic drugs alone for either 4 or 2 days. Importantly, in some cases, when ribociclib treatment preceded a cytotoxic drug, reduced cytotoxicity was observed as compared to two-day treatment with the same cytotoxic drug alone (Fig 1C–1F and S1H–S1J Fig). These results indicate that while concurrent combinations of a CDK4/6 inhibitor with etoposide and paclitaxel in GBM LN308 and lung adenocarcinoma A549 cell lines appeared to produce some additive cytotoxicity, similar to previously being reported [40], sequential schedules did not produce the same additivity. In addition, most other combination schedules, especially the sequential, of ribociclib and a S-phase active (i.e. genotoxic) drug did not increase the genotoxic drug's cytotoxicity. To assess whether these observations were specific to ribociclib or rather a class effect of CDK4/6 inhibitors in general, we determined the $IC_{50}$ of another CDK4/6 inhibitor palbociclib (S2A Fig) and repeated these same experiments combining palbociclib with carmustine or carboplatin in LN428 cells and carboplatin or paclitaxel in A549 cells. Similar to ribociclib, most combinations of palbociclib and these cytotoxic drugs did not lead to enhanced cytotoxicity, and the sequential schedules, especially with palbociclib preceding cytotoxic drugs, diminished cytotoxicity (S2B and S2C Fig). Taken together, these results indicate that the lack of cooperativity, and, in certain sequential combinations, detriment in cancer cell killing when CDK4/6 inhibitors are combined with cytotoxic drugs are likely a class effect of CDK4/6 inhibitors.

We reason that this general lack of cooperativity may either be inherent in the differences in the mechanisms of action of these two classes of drugs or may signal that a more precise scheduling coordination is required to produce such a cooperative effect. To explore these possibilities, we tested whether release of synchronized cells induced by one class of drugs (e.g. G1 or G2/M synchronized cells treated with ribocilicb or cytotoxic drugs, respectively) was necessary for the maximal inhibitory effect of the other drug class, we introduced a 24-hr drug-free interruption in treatment between the two drug classes. Controls were cells treated with a total of 4 days of cytotoxic drugs or vehicle alone with a 24-hr interruption in the middle (Fig 2A). Similar to the uninterrupted schedules, treatment with ribociclib prior to a cytotoxic drug demonstrated only a modest increase in cytotoxicity when compared to vehicle-treated controls, yet significantly reduced cytotoxicity when compared to cytotoxic drugs alone or cytotoxic drugs followed by ribociclib (Fig 2B–2E and S3 Fig). In addition, there was no significant difference in cytotoxicity between 2 days of cytotoxic drug treatment followed by ribociclib and 4 days of cytotoxic drugs. Although this observation may suggest that ribociclib enhances cytotoxicity when given after a cytotoxic drug, it is more likely that the addition of ribociclib

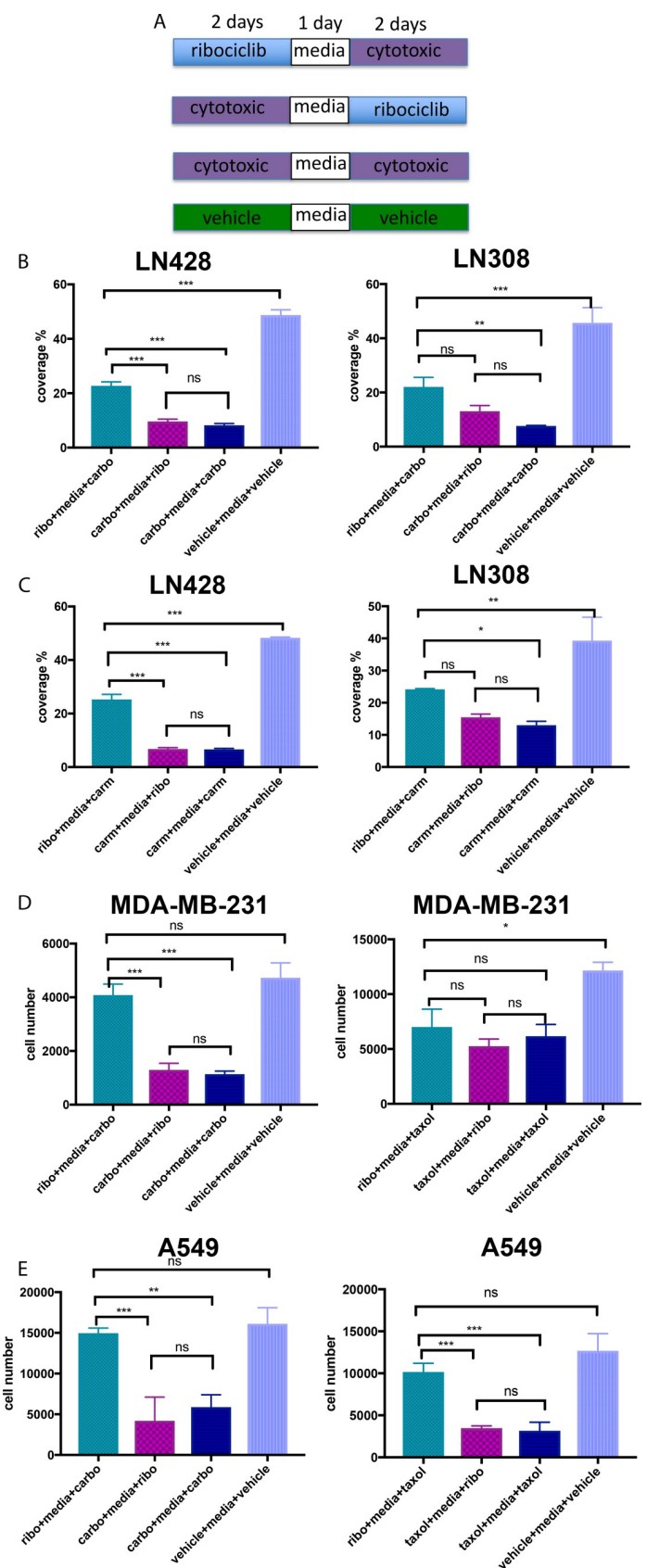

**Fig 2. Interrupted schedules of ribociclib with cytotoxic drugs did not increase cytotoxicity.** (A) Diagrams of interrupted schedules of treatment with ribociclib and cytotoxic drugs. (B-E) Graphs of representative cytotoxicity assay of 3 independent repeats of the various combinations of ribociclib and indicated cytotoxic drugs as shown in A at the IC$_{50}$ concentration for each drug in LN428 (B), LN308 (C), MDA-MB-231 (D) and A549 cells (E). All values are numbers of live cells remaining in culture at the end of treatment and presented as mean (SD). P-value was calculated by one-way ANOVA: *, p<0.033; **, p <0.02; ***, p < 0.001.

did not add to cytotoxicity and 2 days of treatment with a cytotoxic was sufficient at achieving maximal cytotoxicity since 2- and 4-day treatment with a cytotoxic drug alone had similar cytotoxicity (Fig 1C–1F). Taken together, these results indicate that allowing arrested cells time to re-enter the cell cycle did not generate a cooperative condition for additive or synergistic effects when combining CDK4/6 inhibitors with cytotoxic drugs, and further confirm that pre-exposure to CDK4/6 inhibitors may reduce cytotoxicity of cytotoxic drugs.

## Synchronized release of ribociclib-induced G1/S arrested cells into cycling unexpectedly leads to reduced cytotoxicity of cytotoxic drugs

Next, we asked whether synchronized release of ribociclib-induced G1/S arrested cells into cycling may produce the potential synergy with cytotoxic drugs that was not realized by unsynchronized combinations tested above. To achieve synchronized release, we first established the optimal timing of ribociclib-induced G1/S arrest and the kinetics of release from the arrest after ribociclib withdrawal in all four cell lines, using propidium iodine (PI)-based DNA content analysis by flow cytometry (Fig 3 and S4 Fig). The percentage of cells at each stage of the cell cycle was monitored daily before, during and after withdrawal of treatment with ribociclib at its corresponding IC$_{50}$ concentration. Twenty-four hours of exposure to ribociclib was sufficient to achieve maximal arrest at the G1/S checkpoint as demonstrated by the G1 fraction increasing from 48–62.5% before treatment to 87–97% after treatment in these four cell lines (Fig 3B and 3C and S4B and S4C Fig). Unexpectedly, prolonged exposure to ribociclib did not increase the G1 fraction further, but rather promoted a gradual release of arrested cells from the G1/S checkpoint, resulting in an increase in both the S and G2-M fractions and a corresponding reduction in the G1 fraction. Although the escaped population was small and its mechanism is unclear, this phenomenon was observed in all 4 cell lines with similar kinetics (Fig 3B and 3C and S4B and S4C Fig). To minimize the effect of this escape in subsequent experiments, we selected 24 hours of ribociclib treatment as the optimal G1/S arrest synchronization. Next, the kinetics of release from the G1/S arrest was measured after ribociclib withdrawal. Just as in the G1/S arrest synchronization, release from G1/S arrest was also brisk starting by 24 hours after withdrawal (Fig 3D–3F and S1D–S1F Fig). Based on these results, we selected 24-hr treatment with ribociclib followed by 24-hr drug withdrawal as the optimal synchronized release regime and tested whether it could create a synergistic cell killing effect with subsequent exposure to cytotoxic drugs.

Equal numbers of cells were plated overnight and then treated for 24 hours with ribociclib at each cell line-specific IC$_{50}$ dose or vehicle control media. To minimize differences in cell density at the time of treatment with cytotoxic drugs, which may influence sensitivity to cytotoxic effects of these drugs, the same number of ribociblib-induced G1/S arrested and vehicle-treated cells were re-seeded and cultured for 24 hours in ribociclib-free media to initiate synchronized release. This was followed by 72 hours treatment with a cytotoxic drug and live cell numbers determined (Fig 4A and S5A Fig). Although synchronized release from G1/S arrest enhanced cytotoxicity of cytotoxic drugs in LN428 GBM cells (Fig 4B), it failed to replicate in LN308 GBM cells (S5C Fig) and MDA-MB-231 (Fig 4C). In contrast, in A549 cells, synchronized release created increased resistance to cytotoxic drugs (S5B Fig). Taken together,

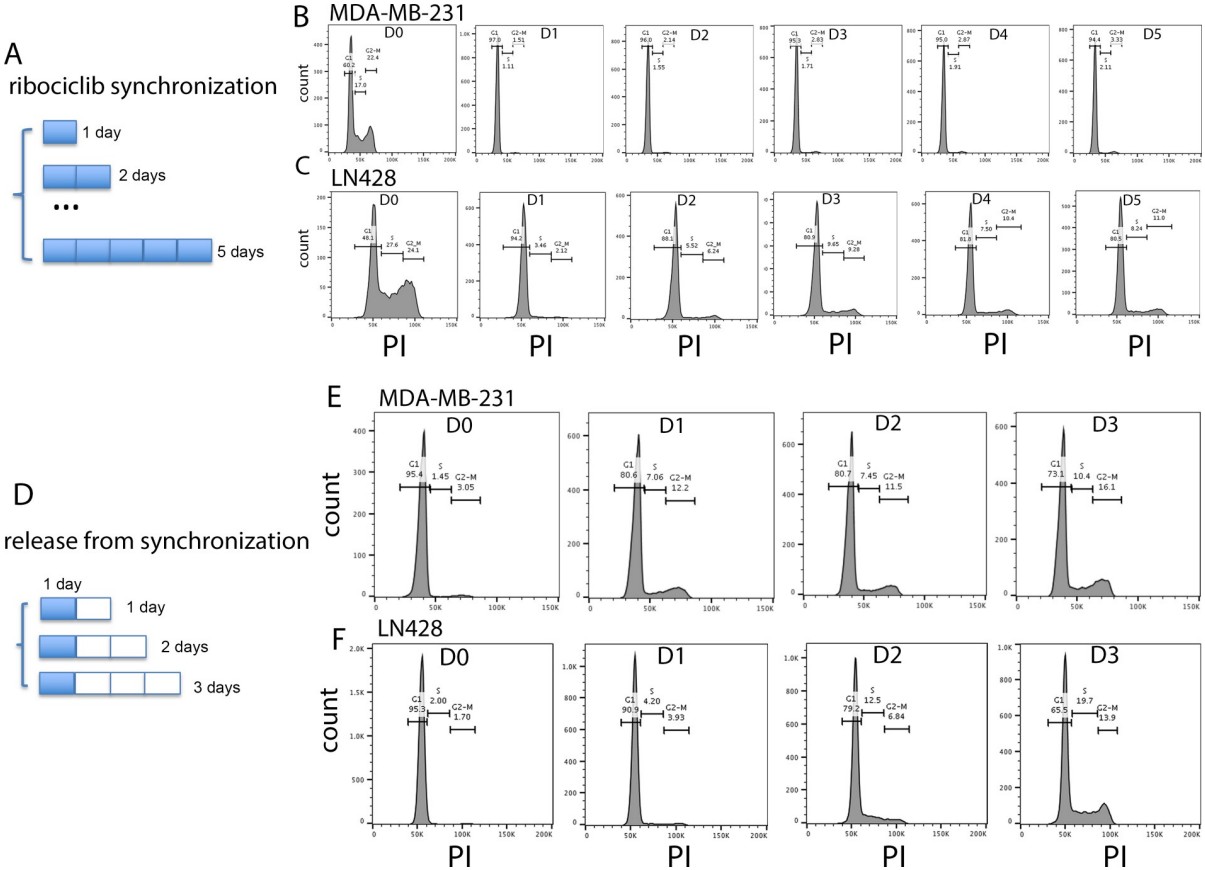

**Fig 3. Optimal synchronization-release regime for ribociclib-induced arrest at the G1/S checkpoint.** (A) A diagram of G1/S synchronization by ribociclib. (B-C) Representative histograms of cell cycle analysis of MDA-MB-231 (B) and LN428 (C) cancer cell lines treated with ribociclib for 0–5 days (D0-D5). Percentages of cells at different stages of the cell cycle are listed. (D) A diagram of release schedule from ribociclib-induced G1/S arrest synchronization.(E-F) Representative histograms of cell cycle analysis of MDA-MB-231 (B) and LN428 (C) cancer cell lines treated with ribociclib for 1 day followed by ribociclib withdrawal for 0–3 days (D0-D3). Percentages of cells at different stages of the cell cycle are listed.

ribociclib-synchronized arrest and release does not consistently improve efficacy of cytotoxic drugs in these cancer cell lines.

As currently approved, each treatment cycle of ribociclib is normally administered daily for at least 3 weeks uninterrupted. However, as demonstrated above (Fig 3B and 3C and S4B and S4C Fig), prolonged treatment with ribociclib led, not to maximal G1/S synchronization, but instead to gradual escape from arrest. Whether this slow release of cells by escape generates cells that remain sensitive to cytotoxic drugs or instead cells with escape mechanisms that can overcome subsequent cell cycle checkpoints is unclear. To address these questions, we treated cells with ribociclib continuously for 5 days, released by 24 hours ribociclib withdrawal, and followed by 3 days of treatment with a cytotoxic drug. Prolonged exposure to ribociclib reversed LN428's sensitivity to cytotoxic drugs observed with a shorter exposure to ribociclib and instead created resistant cells (compare Fig 4B to Fig 4D), while producing no impact on the rest of the other cell lines examined (Fig 4E and S5D and S4E Figs).

For the S-phase active agents examined (etoposide, irinotecan, carboplatin and carmustine), sensitivity occurs when cells progress through DNA replication. It is possible that the 24 hours release from the ribociclib-synchronized arrest indicated above failed to capture many reentry cells during their S-phase transition since there was already 10–20% of cells containing

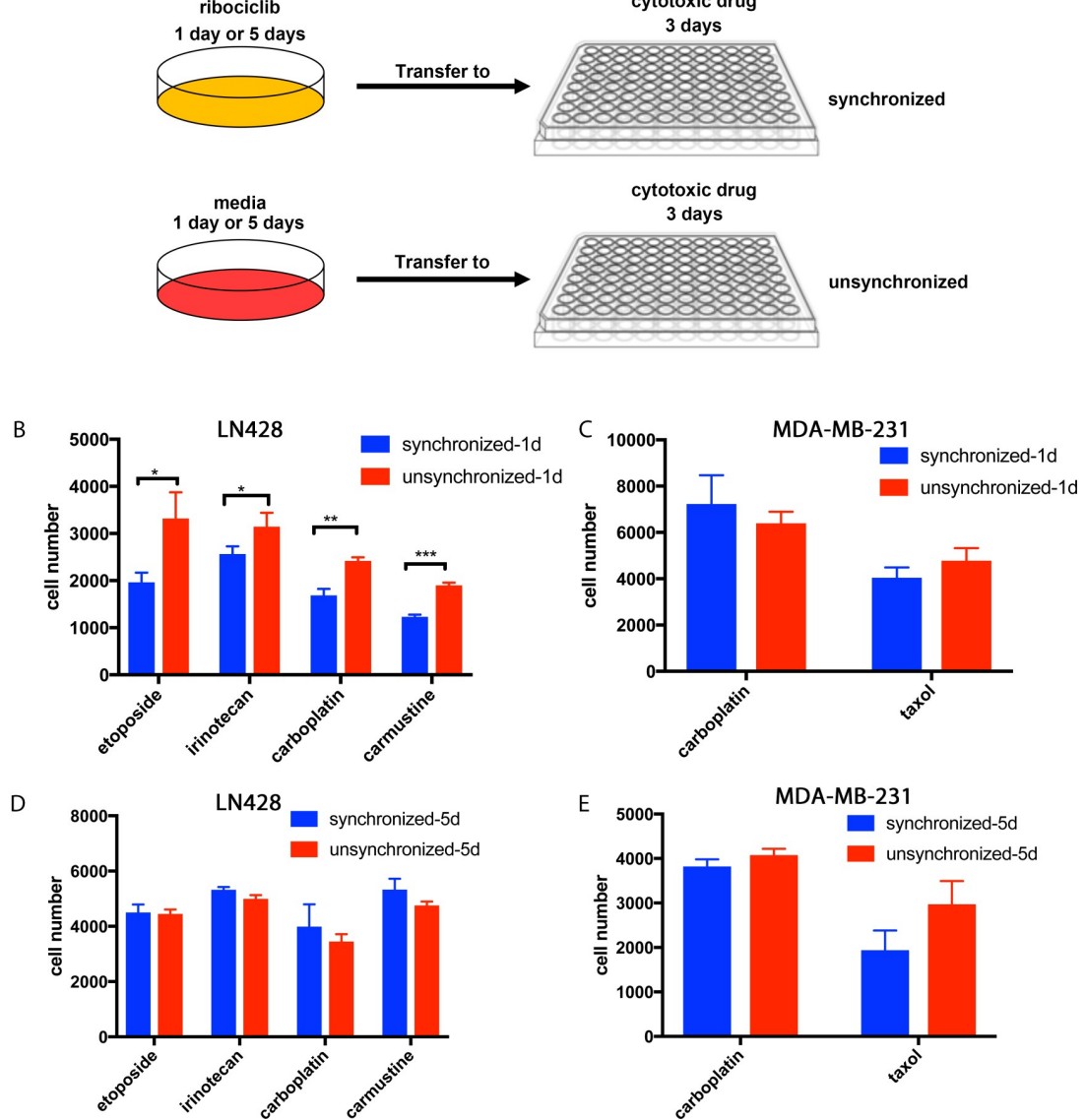

**Fig 4. Synchronized release from ribociclib-induced G1/S checkpoint arrest did not increase cytotoxicity of cytotoxic drugs.**
(A) Diagrams of experimental and control treatment schedule based on the synchronization-release schedules shown in Fig 3.
(B-C) Representative graphs of 3 independent repeats of the cytotoxicity assay in indicated cells treated with indicated cytotoxic drugs after the 1-day synchronization-1-day release regime as shown in A. (D-E) Representative graphs of 3 independent repeats of the cytotoxicity assay in indicated cells treated with indicated cytotoxic drugs after the 5-day synchronization-1-day release regime as shown in A. All values are numbers of live cells remaining in culture at the end of treatment and presented as Mean (SD). P-value was calculated using 2-sided T-test: *, p<0.05; **, p <0.01; ***, p < 0.001.

G2 phase DNA content, which would not be sensitive to S-phase active agents. Therefore, to more accurately ensure correct timing of maximal synchronization of S-phase entry and exposure to S-phase active agents, we monitored S phase entry after release (i.e. ribociclib withdrawal) by measuring timed incorporation of the thymidine analog bromodeoxyuridine (BrdU) into DNA during DNA replication [25]. Cells were treated with ribociclib for 24 hours followed by ribociclib withdrawal in a BrdU containing media, and fractions of BrdU-positive cells determined at 0, 2, and 4 hours after release. In the two cell lines tested, A549 and LN428,

within 2 hours of release, cells have begun to enter S phase as the BrdU-positive fraction increased more than 20 folds (Fig 5A–5C). Released cells were then treated with cytotoxic drugs either at the time of or 2 hours after release. Since the rate of intracellular accumulation of cytotoxic drugs may differ among cell lines, we also tested adding cytotoxic drugs at 6 and 2 hours before release (Fig 5D). Controls were cells treated with cytotoxic drugs for 3 days followed by 1-day treatment with media or ribociclib. There was no improvement in cytotoxic in any of the timed coordination compared to treatment with cytotoxic drugs alone. In fact, adding cytotoxic drugs at the time of or 2 hours after release, actually impaired cytotoxicity of certain cytotoxic drugs (e.g. paclitaxel in A549 cells and carboplatin in LN428 cells) (Fig 5E and 5F).

In summary, we did not detect any significant advantage of inhibiting CDK4/6 before cytotoxic drugs, and prolonged CDK4/6 inhibition did not enhance cytotoxic drug efficacy. And in some cancer cell lines this combination may render cells more resistant to cytotoxic chemotherapy.

## Discussion

Overactive CDK4/6-dependent pathway is frequently observed in many human cancers. Hence, inhibition of CDK4/6 has emerged as an attractive therapeutic strategy against cancer. Based on the mechanism of action, CDK4/6 inhibitors are predicted to have cytostatic property. Therefore, the greatest therapeutic potential of CDK4/6 inhibitors lies in their combinations with other therapies. Combinations of CDK4/6 inhibitor with hormonal therapy in patients with HR+ breast cancer enhanced the inhibition of tumor development and progress free survival in clinical trial [26, 27, 29] and have been approved by FDA for the treatment of metastatic HR+ breast cancer. However, combining CDK4/6 inhibitors with cytotoxic drugs has not been well addressed, with earlier conflicting results. Due to the distinct phases of the cell cycle at which these 2 classes of agents act, it has remained unclear how best to combine them to achieve enhanced cytotoxicity without creating antagonism. Our study evaluated the efficacy of combining CDK4/6 inhibitors with 6 cytotoxic agents widely used for glioblastoma, lung and breast cancers in either concurrent or sequential administration schedules. The six cytotoxic agents used in this study have distinct mechanisms of action. Although alkylating agents (e.g. carmustine, temozolominde in this study) and platinum-based agents (e.g. carboplatin) are not cell cycle dependent, fast dividing cells in general are more sensitive to these agents than are non-dividing cells, suggesting that the cytotoxicity of these drugs are dependent on cell cycling. On the other hand, cell cycle dependent cytotoxic agents exert their cytotoxicity by disrupting cell cycle phase specific functions, thereby forcing associated cell cycle phase checkpoints, e.g. etoposide and irinotecan, S phase-specific topoisomerase inhibitors specifically target DNA replication and the S phase checkpoint, and paclitaxel, a microtubule-stabilizing agent specifically targets the M phase checkpoint. Despite these differences in their mechanisms of action, the results demonstrated that, in all 4 different cancer cell lines tested, combinations of CD4/6 inhibitors, e.g. ribociclib and palbociclib, and the 6 cytotoxic agents, given either concurrently or sequentially failed to consistently produce an additive or synergistic cytotoxic effect. Of note, treating cell with CDK4/6 inhibitors before cytotoxic drugs yielded an antagonistic effect, even when cells were released from ribociclib-induced G1/S synchronized arrest and exposure to a cytotoxic drug was timed to cellular entry into the S phase. These results bring into question the therapeutic utility of combining these 2 classes of drugs regardless of sequence and urge cautions when CDK4/6 inhibitors are administered prior to cytotoxic drugs. The mechanism underlying the antagonism of CDK4/6 inhibitors pre-treatment against cytotoxic drugs, but not when they are given concurrently or after cytotoxic

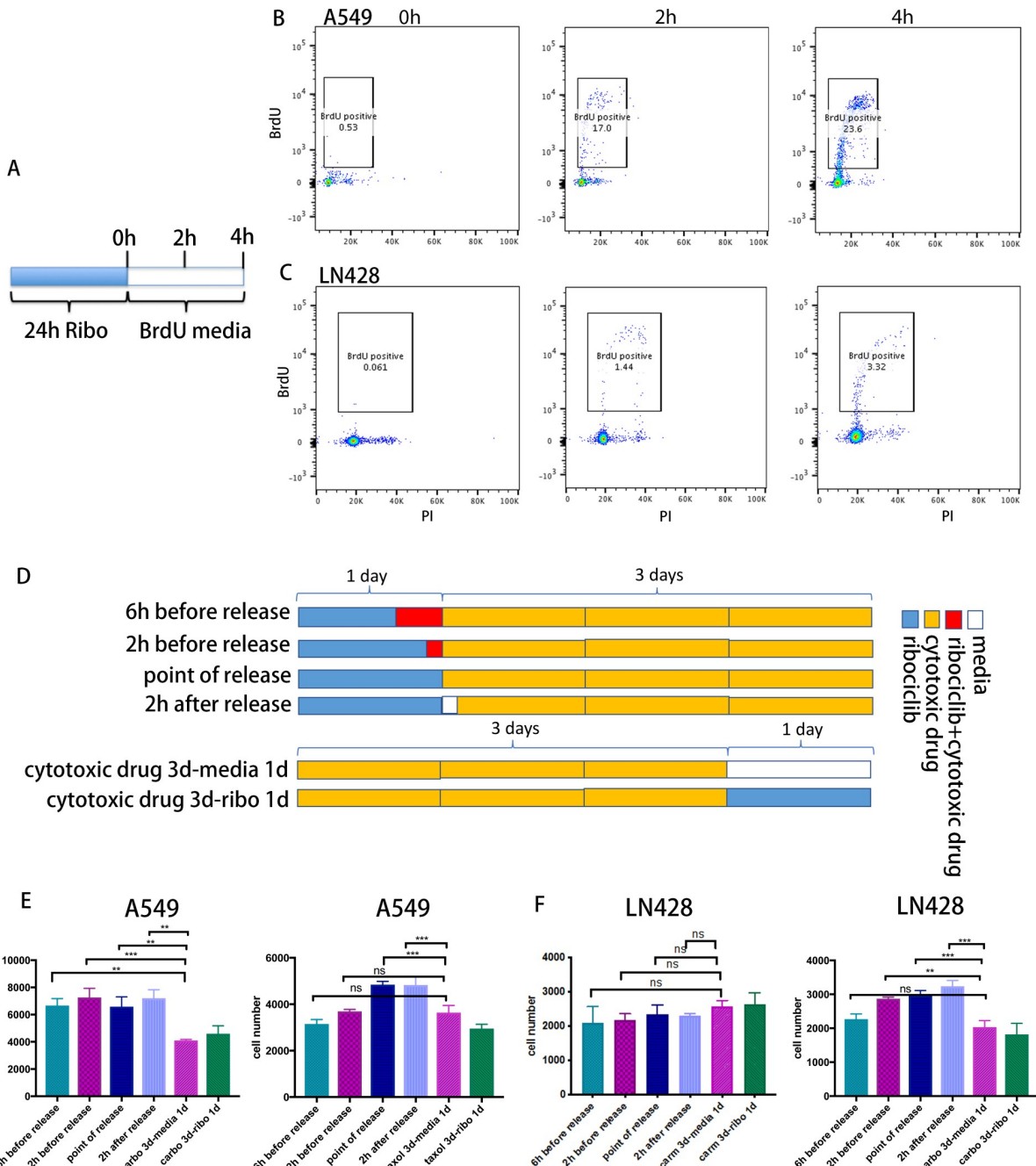

**Fig 5. Precise coordination of S-phase entry after ribociclib withdrawal and exposure to cytotoxic drugs did not enhance cytotoxicity.**
(A) A diagram of the schedule for release from ribociclib-synchronized G1/S arrested cells.(B-C) Representative FACS plots of a BrdU incorporation assay in A549 (B) and LN428 (C) cell lines after release from ribociclib-synchronized G1/S arrest. (D) Diagrams of coordination schedule of S-phase entry (BrdU incorporation) and treatment with cytotoxic drugs.(E-F) Representative graphs of 3 independent repeats of a cytotoxicity assay based on the coordination schedule in D in A549 (E) and LN428 (F) cell lines. carbo: carboplatin; carm: carmustine. All values are numbers of live cells remaining in culture at the end of treatment and presented as mean (SD). P-value was calculated by one-way ANOVA: *, $p < 0.033$; **, $p < 0.02$; ***, $p < 0.001$.

drugs is unclear and could be due to the observation that CDK4/6 inhibitors shifted the burden of E2F-induced DNA repair from homologous recombination to non-homologous end joining, leading to antagonism with cytotoxic agents and contribute to increased growth [30].

Alternatively, cell cycle arrest at $G_0/G_1$ transition caused by CDK4/6 inhibitors may nullify cell division dependent cytotoxicity of cytotoxic agents. Recently, senescence induced by CDK4/6 inhibition has also been shown to promote the development of cancer stem-like cells [41–44], leading to attenuated response to cytotoxic agents and tumor recurrence. The possibility that CDK4/6 inhibitors may antagonize cytotoxic drugs though interfering with transport systems such the ATP-binding cassette (ABC) transporters system to reduce intracellular delivery of cytotoxic drugs leading to multidrug resistance development is unlikely to account for the observed antagonism because recent studies demonstrated that many CDK4/6 inhibitors including ribociclib and palbociclib were potent inhibitors of several ABC transporters [45–47]. Furthermore, the observation that more prolonged exposure to ribociclib led to escape from the G1/S arrest and further nullified cytotoxicity of cytotoxic drugs has practical clinical implications as ribociclib, like other approved CDK4/6 inhibitors, are routinely prescribed for weeks at a time. The mechanism of this escape is unclear but may be due in part to compensatory upregulation of CDK6 and cyclin E or via cyclin D1-CDK2-mediated escape [48–50].

The main limitation of our study lies in its in vitro setting. Whether the same observations hold true in vivo remains to be investigated. Until such study is completed, extra precautions are warranted when these classes of drugs are combined in the clinic.

## Supporting information

**S1 Fig. Combining ribociclib with cytotoxic drugs did not increase cytotoxicity.** (A-G) Representative dose-response curves of 3 independent biological repeats of ribociclib (A), carboplatin (B), carmustine (C), paclitaxel (Taxol) (D), temozolomide (TMZ) (E), etoposide (F) and irinotecan (G) for the growth inhibition of LN428, LN308, A549 and MDA-MB-231 are shown. Each data point was done in triplicates. (H-J) Graphs of representative cytotoxicity assay of 3 independent repeats of the various combinations of ribociclib as shown in Fig 1B at the $IC_{50}$ concentration for each drug in LN428 and LN308 cells with TMX (H), etoposide (I) and irinotecan (J). ribo: ribociclib; TMZ: temozolomide; eto: etoposide; iri: irinotecan. All values are numbers of live cells remaining in culture at the end of treatment and presented as mean (SD). P-value was calculated by one way ANOVA: *, $p < 0.033$; **, $p < 0.02$; ***, $p < 0.001$.
(TIF)

**S2 Fig. Combining palbociclib with cytotoxic drugs did not increase cytotoxicity.** (A) Representative dose-response curves of 3 independent biological repeats of palbociclib in LN428 and A549 cells are shown. Each data point was done in triplicates. (B-C) Graphs of representative cytotoxicity assay of 3 independent repeats of the various combinations of palbociclib at its $IC_{50}$ concentration in LN428 (B) and A549 (C) cells with indicated cytotoxic drugs. palbo: palbociclib; carm: carmustine; carbo: carboplatin. All values are numbers of live cells remaining in culture at the end of treatment and presented as mean (SD). P-value was calculated by one way ANOVA: *, $p < 0.033$; **, $p < 0.02$; ***, $p < 0.001$.
(TIF)

**S3 Fig. Interrupted schedules of ribociclib with cytotoxic drugs did not increase cytotoxicity.** (A-B) Graphs of representative cytotoxicity assay of 3 independent repeats of the various combinations of ribociclib and indicated cytotoxic drugs as shown in Fig 2A at the $IC_{50}$ concentration for each drug in LN428 (A) and LN308 (B) cells (E). All values are numbers of live cells remaining in culture at the end of treatment and presented as mean (SD). P-value was calculated by one way ANOVA: *, $p < 0.033$; **, $p < 0.02$; ***, $p < 0.001$.
(TIF)

**S4 Fig. Optimal synchronization-release regime for ribociclib-induced arrest at the G1/S checkpoint.** (A) A diagram of G1/S synchronization by ribociclib. (B-C). Representative histograms of cell cycle analysis of A549 (B) and LN308 (C) cancer cell lines treated with ribociclib for 0–5 days (D0-D5). Percentages of cells at different stages of the cell cycle are listed. (D) A diagram of release schedule from ribociclib-induced G1/S arrest synchronization. (E-F) Representative histograms of cell cycle analysis of A549 (B) and LN308 (C) cancer cell lines treated with ribociclib for 1 day followed by ribociclib withdrawal for 0–3 days (D0-D3). Percentages of cells at different stages of the cell cycle are listed.
(TIF)

**S5 Fig. Synchronized release from ribociclib-induced G1/S checkpoint arrest did not increase cytotoxicity of cytotoxic drugs.** (A) Diagrams of experimental and control treatment schedule based on the synchronization-release schedules shown in Fig 3. (B-C) Representative graphs of 3 independent repeats of the cytotoxicity assay in indicated cells treated with indicated cytotoxic drugs after the 1-day synchronization-1-day release regime as shown in A. (D-E) Representative graphs of 3 independent repeats of the cytotoxicity assay in indicated cells treated with indicated cytotoxic drugs after the 5-day synchronization-1-day release regime as shown in A. All values are numbers of live cells remaining in culture at the end of treatment and presented as Mean (SD). P-value was calculated using 2-sided T-test: *, $p < 0.05$; **, $p < 0.01$; ***, $p < 0.001$.
(TIF)

## Acknowledgments

We thank members of our laboratory for their technical assistance and helpful comments.

## Author Contributions

**Conceptualization:** Dan Jin, Nguyen Tran, David D. Tran.

**Data curation:** Dan Jin, Nguyen Tran, Nagheme Thomas.

**Formal analysis:** Dan Jin, Nguyen Tran, David D. Tran.

**Funding acquisition:** David D. Tran.

**Investigation:** Dan Jin, Nguyen Tran, David D. Tran.

**Methodology:** Dan Jin, Nguyen Tran, Nagheme Thomas, David D. Tran.

**Project administration:** David D. Tran.

**Resources:** Nagheme Thomas, David D. Tran.

**Supervision:** David D. Tran.

**Validation:** Dan Jin, Nguyen Tran.

**Writing – original draft:** Dan Jin, Nguyen Tran, David D. Tran.

**Writing – review & editing:** Dan Jin, Nguyen Tran, Nagheme Thomas, David D. Tran.

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
