## [Decision Letter · Decision Letter 0]

6 Jul 2019

PONE-D-19-15272

Combining a CDK4/6 inhibitor with Cytotoxic Agents Does Not Enhance Cytotoxicity

PLOS ONE

Dear Dr. Tran,

Thank you for submitting your manuscript to PLOS ONE. After careful consideration, we feel that it has merit but does not fully meet PLOS ONE’s publication criteria as it currently stands. Therefore, we invite you to submit a revised version of the manuscript that addresses the points raised during the review process.

As noted by the reviewer's comments below, a number of clarifications are requested including contributions of specific cell cycle phase in synergistic/cytotoxic effects of the agents utilized in this study, and a different perhaps additional method of statistical analyses for some data. Other minor issues include a more focussed title and a revision of the discussion section.

We would appreciate receiving your revised manuscript by Aug 20 2019 11:59PM. To enhance the reproducibility of your results, we recommend that if applicable you deposit your laboratory protocols in protocols.io, where a protocol can be assigned its own identifier (DOI) such that it can be cited independently in the future. For instructions see: http://journals.plos.org/plosone/s/submission-guidelines#loc-laboratory-protocols

We look forward to receiving your revised manuscript.

Kind regards,

Arun Rishi, Ph.D.

Academic Editor

PLOS ONE

Journal Requirements:

This work was supported by grants from Novartis, Inc. and the National Institutes of Health (D.D.T). The funders had no role in the study design, data collection and analysis, decision to publish or preparation of the manuscript.

We note that you received funding from a commercial source: Novartis, Inc.

Reviewers' comments:

Reviewer's Responses to Questions

**Comments to the Author**

1. Is the manuscript technically sound, and do the data support the conclusions?

Reviewer #1: Partly

Reviewer #2: Partly

2. Has the statistical analysis been performed appropriately and rigorously? 

Reviewer #1: No

Reviewer #2: Yes

3. Have the authors made all data underlying the findings in their manuscript fully available?

Reviewer #1: Yes

Reviewer #2: Yes

4. Is the manuscript presented in an intelligible fashion and written in standard English?

Reviewer #1: Yes

Reviewer #2: Yes

5. Review Comments to the Author

Reviewer #1: This is in general an interesting study evaluating the apoptotic/cytotoxic effect of CDKI4/6 inhibitors when combined with conventional cytotoxic drugs. Nevertheless, the evaluation of the results should be done properly as well as their interpretation. In fact the authors more or less repeat the statements from introduction and results also in the discussion not bringing too much light into this problematics. I have few concerns to the study as stated in major comments, which might help improve the takehome message of the study when addressed properly.

Major comments:

Before driving the conclusion that the CDKI4/6 inhibitors should not be combined with anticancer drugs in clinical settings, other mechanisms of possible synergism must be considered. Authors do few attempts to address the issue in the discussion, 472-489, nevertheless, the hypothesis on ABC transporters (lines 477-480) is wrong. There is no need to consider ABC efflux transporters to represent a mechanism for antagonism in this case! The authors correctly mention that several previous studies revealed first/second generation of CDKI as potent inhibitors of ABC efflux transporters, which are commonly expressed in cancer cells (citing references 41-43). Nevertheless, they completely ignore the content of these articles, e.g. the recent study of Sorf et al., 2018 clearly show the synergistic cytotoxic and proapoptotic effect of combination of ribociclib with mitoxantrone and daunorubicin in ABC-expressing cells. This factor must be discussed, even though, it did not play probably role in the results of this study since the cell lines used (such as MDA-MB-231) are known for only negligible and non-physiological levels of ABC transporters. So it is very probably, that in the clinical conditions this effect of CDKI on ABC transporter-mediated efflux of cytotoxic drugs that results in synergism in cytotoxicity might overwhelm the antagonism described here. This must be considered and at least discussed properly!

Introduction lines 97-99 the authors mention „general dependency of many cytotoxic drugs on cell division“. This statement is only partly true, since e.g. alkylating agents, such as cisplatin are able to act also on quescent cells in G0 Phase. In your study you use several cytotoxic drugs, however their cell cycle phase specificity (or unspecificity) must be considered and the experimental results discussed in this context!

The statistical analysis is completely missing in the methodological part – this must be filled properly. Re-consider also the statistical method used in evaluation of the data shown in Fig. 1 and 2. When comparing for example the effect of added ribociclib on the cisplatin alone the effect of sole ribociclib must be considered and thereby ANOVA should be better used

Minor comments

Page 17 lines 222-223 „…breast and lung and breast cancer cells“ („breast“ is doubled here)

Fig 1 A consistent units should be used in describing the concentration of applied drugs (preferring molarity)

Reviewer #2: In this study, the authors tested various combinations of highly selective and potent CDK4/6 inhibitors with commonly used cytotoxic drugs in several cancer cell lines derived from lung, breast and brain cancers, for their cell-killing effects as compared to monotherapy. They concluded that all combinations, either concurrent or sequential, failed to enhance cell-killing effects, and in certain schedules, pre-treatment with a CDK4/6 inhibitor, densensitized cells to cytotoxic agents. These findings urge cautions when combining these two classes of agents in clinical settings.

Major comments

1. Conflicting results regarding the benefit of combining small molecule CDK4/6 inhibitors with standard cytotoxic chemotherapy have been reported. Results presented in this study may not be a general phenomenon for all of the small molecule CDK4/6 inhibitors and all of the cytotoxic agents. Therefore, the title should at least indicate the specific CDK4/6 inhibitor, ribociclib, used in this study.

2. While the results are interesting and important, however, they were conducted only in cell culture model. As there are major differences between cell culture model and preclinical animal models, it would be helpful if the authors could validate the results from cell culture study by conducting at least one preclinical animal study to demonstrate the major point of the research.

6. PLOS authors have the option to publish the peer review history of their article (what does this mean?). If published, this will include your full peer review and any attached files.

Reviewer #1: No

Reviewer #2: No

---

## [Author Response · Author response to Decision Letter 0]

21 Aug 2019

REVIEWER # 1

Major comments:

1. Before driving the conclusion that the CDKI4/6 inhibitors should not be combined with anticancer drugs in clinical settings, other mechanisms of possible synergism must be considered. Authors do few attempts to address the issue in the discussion, 472-489, nevertheless, the hypothesis on ABC transporters (lines 477-480) is wrong. There is no need to consider ABC efflux transporters to represent a mechanism for antagonism in this case! The authors correctly mention that several previous studies revealed first/second generation of CDKI as potent inhibitors of ABC efflux transporters, which are commonly expressed in cancer cells (citing references 41-43). Nevertheless, they completely ignore the content of these articles, e.g. the recent study of Sorf et al., 2018 clearly show the synergistic cytotoxic and proapoptotic effect of combination of ribociclib with mitoxantrone and daunorubicin in ABC-expressing cells. This factor must be discussed, even though, it did not play probably role in the results of this study since the cell lines used (such as MDA-MB-231) are known for only negligible and non-physiological levels of ABC transporters. So it is very probably, that in the clinical conditions this effect of CDKI on ABC transporter-mediated efflux of cytotoxic drugs that results in synergism in cytotoxicity might overwhelm the antagonism described here. This must be considered and at least discussed properly!

Our response: We have re-phrased this section of the discussion to add other potential mechanisms and for better clarification of the ABC transporter connection (Lines 492-503). 

2. Introduction lines 97-99 the authors mention „general dependency of many cytotoxic drugs on cell division “. This statement is only partly true, since e.g. alkylating agents, such as cisplatin are able to act also on quescent cells in G0 Phase. In your study you use several cytotoxic drugs, however their cell cycle phase specificity (or unspecificity) must be considered and the experimental results discussed in this context!

Our response: We change the statement “general dependency of many cytotoxic drugs on cell division” (lines 96-98) to “This observation was in part expected due to the positive correlation between cytotoxicity of many cytotoxic drugs and the rate of cell division.”

We change the statement “The different dependencies on cell cycle phases for cytotoxic drugs are discussed” (lines 465-476) to “The six cytotoxic agents used in this study have distinct mechanisms of action. Although alkylating agents (e.g. carmustine, temozolomide in this study) and platinum-based agents (e.g. carboplatin) are not cell cycle dependent, fast dividing cells in general are more sensitive to these agents than are non-dividing cells, suggesting that the cytotoxicity of these drugs are dependent on cell cycling. On the other hand, cell cycle dependent cytotoxic agents exert their cytotoxicity by disrupting cell cycle phase specific functions, thereby forcing associated cell cycle phase checkpoints, e.g. etoposide and irinotecan, S phase-specific topoisomerase inhibitors specifically target DNA replication and the S phase checkpoint, and paclitaxel, a microtubule-stabilizing agent specifically targets the M phase checkpoint. Despite these differences in their mechanisms of action, …” 

3.The statistical analysis is completely missing in the methodological part – this must be filled properly. Re-consider also the statistical method used in evaluation of the data shown in Fig. 1 and 2. When comparing for example the effect of added ribociclib on the cisplatin alone the effect of sole ribociclib must be considered and thereby ANOVA should be better used.

Our response: We added the method description for statistical analysis used in this manuscript (Line 157-161) and also re-analyzed p-values by ANOVA for Figures 1, 2, 5, S1, S2, S3. Comparison between different treatments and ribociclib alone was also added in the graphs where appropriate.

Minor comments:

Page 17 lines 222-223 „…breast and lung and breast cancer cells “(„breast“ is doubled here)

Our response: Corrected in line 226

Fig 1 A consistent units should be used in describing the concentration of applied drugs (preferring molarity)

Our response: A new IC50 table has replaced the previous version with consistent molarity concentration units (Figure 1A).

REVIEWER # 2 

Major comments:

1. Conflicting results regarding the benefit of combining small molecule CDK4/6 inhibitors with standard cytotoxic chemotherapy have been reported. Results presented in this study may not be a general phenomenon for all of the small molecule CDK4/6 inhibitors and all of the cytotoxic agents. Therefore, the title should at least indicate the specific CDK4/6 inhibitor, ribociclib, used in this study.

Our response: Although we did not test all of the available CDK4/6 inhibitors, we did test 2 CDK4/6 inhibitors, ribociclib in all the experiments shown and palbociclib in two cell lines LN428 and A549 (Figure S2). The results for palbociclib are consistent with those for ribociclib. Therefore, we have changed to title to specify ribociclib and palbociclib.

2. While the results are interesting and important, however, they were conducted only in cell culture model. As there are major differences between cell culture model and preclinical animal models, it would be helpful if the authors could validate the results from cell culture study by conducting at least one preclinical animal study to demonstrate the major point of the research.

Our response: We appreciate the suggestion and agree that this is an important next step. We did acknowledge the limitation of our study due to its in vitro nature. However, conducting a preclinical animal model study properly will require both time and resources to first optimize pharmacokinetics and pharmacodynamics profiles of individual drugs and then in combinations in a breast cancer model before definitive experiments can proceed. This study is currently in planning, we anticipate at least 1-2 more years to complete. However, as we have pointed out in the manuscript, due to the nature of our observations with direct clinical implications regarding patient safety, we feel that it is important to publish the data in its current form to alert the field that extra precautions may be warranted when these two classes of drugs are combined in the clinic until more definitive preclinical confirmation can be obtained.

---

## [Editor Report · Decision Letter 1]

25 Sep 2019

Combining CDK4/6 Inhibitors Ribociclib and Palbociclib with Cytotoxic Agents Does Not Enhance Cytotoxicity

PONE-D-19-15272R1

Dear Dr. Tran,

We are pleased to inform you that your manuscript has been judged scientifically suitable for publication and will be formally accepted for publication once it complies with all outstanding technical requirements.

With kind regards,

Arun Rishi, Ph.D.

Academic Editor

PLOS ONE
---

## [Editor Report · Acceptance letter]

2 Oct 2019

PONE-D-19-15272R1 

Combining CDK4/6 Inhibitors Ribociclib and Palbociclib with Cytotoxic Agents Does Not Enhance Cytotoxicity 

Dear Dr. Tran:

I am pleased to inform you that your manuscript has been deemed suitable for publication in PLOS ONE. Congratulations! Your manuscript is now with our production department. 

With kind regards,

on behalf of

Prof Arun Rishi 

Academic Editor

PLOS ONE